# Awake Bruxism—Single-Point Self-Report versus Ecological Momentary Assessment

**DOI:** 10.3390/jcm10081699

**Published:** 2021-04-15

**Authors:** Alona Emodi-Perlman, Daniele Manfredini, Tamar Shalev, Ilanit Yevdayev, Pessia Frideman-Rubin, Alessandro Bracci, Orit Arnias-Winocur, Ilana Eli

**Affiliations:** 1Department of Oral Rehabilitation, The Maurice and Gabriella School of Dental Medicine, Tel Aviv University, Tel Aviv 6139001, Israel; tamarashalev@gmail.com (T.S.); ilanit.yev@gmail.com (I.Y.); pessia80@gmail.com (P.F.-R.); elilana@tauex.tau.ac.il (I.E.); 2School of Dentistry, University of Siena, 53100 Siena, Italy; daniele.manfredini75@gmail.com; 3School of Dentistry, University of Padova, 35122 Padova, Italy; info@alessandrobracci.com; 4Department of Oral Pathology, Oral Medicine and Maxillofacial Imaging, The Maurice and Gabriela Goldshleger School of Dental Medicine, Tel Aviv University, Tel Aviv 6139001, Israel; orit_winocur@yahoo.com

**Keywords:** awake bruxism, self-report, ecological momentary assessment, smartphone application

## Abstract

Assessment of awake bruxism (AB) is problematic due to the inability to use continuous recordings during daytime activities. Recently, a new semi-instrumental approach was suggested, namely, ecological momentary assessment (EMA), via the use of a smartphone application. With the application, subjects are requested to report, at least 12 times per day, the status of their masticatory muscle activity (relaxed muscles, muscle bracing without tooth contact, teeth contact, teeth clenching, or teeth grinding). The aim of the present study was to test the association between a single observation point self-report and EMA assessment of AB. The most frequent condition recorded by the EMA was relaxed muscles (ca. 60%) and the least frequent was teeth grinding (less than 1%). The relaxed muscle condition also showed the lowest coefficient of variance over a seven-day period of report. Additionally, only the relaxed muscles and the muscle bracing conditions presented an acceptable ability to assess AB-positive and AB-negative subjects, as defined by single-point self-report questions. The combination between self-report and EMA may have the potential to promote our ability to assess AB. We suggest to re-consider the conditions of teeth contact and teeth grinding while using EMA to evaluate AB.

## 1. Introduction

The definition of awake bruxism (AB) was set by an international bruxism expert panel as a “masticatory muscle activity during wakefulness that is characterized by repetitive sustained tooth contact and/or bracing or thrusting of the mandible and is not a movement disorder in otherwise healthy individuals” [1].

For a recent review on the current knowledge of the etiology, assessment, and management of bruxism, see Manferdini et al. [2]. Concisely, it can be summarized that the exact etiology of bruxism has not been determined yet. Part of bruxism activity is genetically determined, but the inheritance model or genetic markers are unknown [3]. Increasing evidence suggests a combination of psychosocial aspects, especially stress sensitivity, anxiety, and poor coping skills, to be mainly related to AB [4,5,6,7].

The international expert panel suggested grading the assessment of AB as: (i) Possible AB, based merely on a positive self-report; (ii) probable AB, based on a positive clinical inspection, with or without a positive self-report; (iii) definite AB, based on a positive instrumental assessment, with or without a positive self-report and/or a positive clinical inspection [1].

Some studies have suggested that the assessment accuracy of self-reported bruxism is low due to a lack of individual awareness about such a behavior [8,9]. The assessment of probable AB may also be problematic, since the relevant intra-oral signs can also be present in subjects who perform sleep bruxism (SB) [10,11,12,13,14], as well as other oro-motor activities [15]. While both self-report and clinical assessment present some degrees of diagnostic sensitivity, they are insufficient in determining outcomes such as the intensity and duration of a specific muscle activity and its fluctuations over time [16,17]. This leaves instrumental electromyography (EMG) recordings, as the gold standard for the assessment of definite AB. Obviously, a continuous EMG evaluation of muscular activity during daytime is problematic for technical, emotional, and feasibility reasons.

Recently, a new semi-instrumental approach was suggested, namely, ecological momentary assessment (EMA), which was initially developed in response to the limitations of retrospective recall. EMA involves repeated sampling of subject’s current behaviors and experiences and refers to the real-time reporting of a behavior, a feeling, or whichever condition is under study. Individuals are given instructions to record responses whenever they feel a particular way or at given predetermined intervals [17]. EMA that relies on a smartphone application alerts the user at times determined by the evaluator (preset and random). When the user is alerted, prompts are provided on the phone to complete the assessment questions, with the resultant data transmitted to the evaluator [18]. A recent guide to evaluating digital health products summarizes EMA’s pros as reducing the problem of users having to remember how they felt or what they did (recall bias), being an efficient and easy way to collect data and often reducing the number of participants needed for the study, as each participant provides data several times. The drawbacks of EMA include the possibility of being annoying for participants, the possibility of significant missing data and the need for specific skills to analyze the data [19].

With the aim of assessing AB through EMA, a designated smartphone application was developed as a semi-instrumental option for multiple-point real-time subjective report about masticatory muscle activity during wakefulness [20,21,22,23]. Recent studies presented frequencies of the different conditions recorded by the application among healthy young adults (university students) [20,22,23]. One of the conclusions was that the application can mainly be used to assess the frequency of AB behavior and to implement the control of AB in patients with potential clinical consequences [22].

While semi-instrumental EMA via smartphones is gaining popularity as a possible tool to implement the self-reported approach to AB assessment, questionnaires based on a single observation point are still the primary tool to gather data on AB in research and clinical practice [1].

The aim of the present study was to look for associations between two modes of AB assessment—single-point self-report and EMA with the use of a designated smartphone application.

## 2. Materials and Methods

### 2.1. Population

Two hundred and thirty-eight dental students attending the School of Dental Medicine, Tel Aviv University, were approached in January 2020 and requested to respond to self-report questionnaires and to use an EMA application for at least seven consecutive days. One hundred and forty-seven students consented to participate (62% response rate).

### 2.2. Tools

#### 2.2.1. Single-Point Observation Self-Report of AB

An accepted way to assess AB is through a single-point observation self-report [24,25,26,27,28,29]. In the present study, participants were requested to respond to three questions related to their awareness to grinding, clenching, holding the teeth together, and/or tightening the masticatory muscles during the day as follows:(i)Do you grind your teeth during the day (namely, do you move repeatedly your jaw from one side to another and/or forward and backwards with teeth contact)?—*Self report grinding.*(ii)Do you clench your teeth during the day (namely, do your lower teeth touch your upper teeth, even if lightly)?—*Self-report teeth contact/clenching.*(iii)Do you experience muscle bracing during the day (namely, are your jaw muscles strained, kept in a fixed position without teeth contact)?—*Self-report bracing*.

The scoring possibilities for each of the questions were as proposed by the Diagnostic Criteria for Temporomandibular Disorders (DC/TMD) oral behavior checklist [30]: 0 = never, 1 = almost never, 2 = some of the time, 3 = most of the time, and 4 = all of the time.

A score of 2, 3, or 4 on one or more of the three questions was considered as positive AB.

The questionnaire included additional information concerning demographics and variables such as oral parafunctions, anxiety, depression, temporomandibular disorders (TMDs) and additional clinical information, which are not part of the present study.

#### 2.2.2. EMA with the Use of a Designated Smartphone Application

The BruxApp is an EMA application, designated for AB assessment. The use of BruxApp was as described in detail by Bracci et al. [20]. In the present study, we followed the protocol described by Bracci et al., with an officially translated Hebrew version of the application.

In brief, participants received two explanation/training sessions of how to use the application and how to recognize the different conditions by one of the investigators (T.S.-A.). Following the explanation, they downloaded the application to their smartphones. The BruxApp sent each participant 20 alert sounds at random hours during the day. At each alert sound, the subject was requested to indicate on his/her smartphone (within 5 min from the alert sound), the present condition of his/her teeth and jaw position as follows:(i)Relaxed jaw muscles: Condition of perceived relaxed state of jaw muscles, with jaws kept apart (BA (BruxApp)-relaxed).(ii)Muscle bracing (without tooth contact): Condition of jaw muscle stiffness or tension, as in tooth clenching but with the teeth apart (BA-bracing).(iii)Teeth contact: Condition of slight tooth contact, as if a 40 µ articulating paper is placed between the dental arches, and the subject is asked to keep it there by lightly touching the teeth together with the mouth closed (BA-teeth contact). The assumption was that participants (all dental students) were well familiar with the articulating paper procedure and that the example would help them to better discriminate between teeth contact and the condition teeth clenching (see below).(iv)Teeth clenching: All conditions where tooth contact is more marked than those listed above, and the jaw muscles are tensed (BA-clenching).(v)Teeth grinding: Condition in which the patient gnashes or grinds the opposing teeth, independent of the intensity and direction of antagonist tooth contact (BA-grinding).

The software (BruxApp, Pontendra, Italy) was programmed to send 20 alerts during the day at random intervals to limit expectation bias (e.g., the risk that individuals may modify their behaviors based on the alert expectation, if set at predetermined intervals). Recording time was set starting 1 h. post-awakening in the morning, for 12 h. Data were recorded over a seven-day period. Colonna et al. studied subjects’ compliance with the software and the 12 out of 20 alert threshold showed the best available results [23]. Therefore, a minimum of 12 responded alerts per day was required to be included in the analysis. Days with less than 12 responded alerts were discarded. Subjects who failed to complete seven days of valid responses were not included in the study.

The study was approved by the ethical committee of Tel-Aviv University, approval no. 000693-1. Written informed consent was obtained from all the participants.

### 2.3. Statistics

Mean number of reports of each of the BruxApp conditions per day was calculated for the study population, as well for the sum number of reports of four conditions other than BA-relaxed (BA-bracing, BA-teeth contact, BA-clenching, and BA-grinding) per day (BA-awake bruxism, BA-AB).

The frequencies (percentage) of each of the BruxApp conditions were calculated as described by Bracci et al. [20]. The frequency of each condition (BA-relaxed, BA-bracing, BA-teeth contact, BA-clenching, and BA-grinding) was calculated as a percentage with respect to the answered alerts for all individuals. The frequencies were calculated daily, on an individual basis, and individual frequencies were used to calculate an average of the study population on a daily basis. Data were reported as a range of mean values of the seven-day span, per each condition. For each condition, a coefficient of variation (CV) of the frequency data was assessed.

Pearson’s Chi-square test and Fisher’s exact test were used to test the associations between categorical variables.

The area under the receiver operating characteristic curve (ROC-AUC) was used to evaluate the ability of the BruxApp variables to discriminate between positive and negative AB, as defined by single-point self-report.

## 3. Results

### 3.1. Descriptive Results

Of the initial 147 dental students who consented to participate in the study, 106 (67 female) completed seven days of a full BruxApp response (at least 12 responses/day, minimum of 84 responses) and were included in the final analysis.

The mean age of the study population was 24.4 ± 2.99 years. Of the study population, 36 subjects (34%) were defined as AB-positive according to single-point self-report (no differences between genders).

The total number of BruxApp reports for the entire study population, over the seven days period of observation, was 11,122. The mean number of reports per day, for each of the BruxApp conditions, was as follows: BA-relaxed: 8.96 ± 4.59; BA-bracing: 2.05 ± 2.62; BA-teeth contact: 3.04 ± 3.0; BA-clenching: 0.88 ± 2.02; BA-grinding: 0.08 ± 0.42.

### 3.2. Frequencies of the BruxApp Behaviors

The mean frequencies for each of the BruxApp conditions per day, over the seven-day observation period, are presented in Figure 1. The mean frequencies of the different BruxApp conditions, over the seven report days, ranged as follows: BA-relaxed: 55.5–63.7%; BA-bracing: 11.6–19.8%; BA-teeth contact: 17.9–21.7%; BA-clenching: 5–6.5%; BA-grinding: 0.4–0.9%. The lowest CV was observed for the condition BA-relaxed (0.48) and the highest for the condition BA-grinding (5.19).

### 3.3. Correlations between Single-Point Self-Report AB and the BruxApp Conditions

Pearson’s correlation coefficient was computed between possible AB as defined by single-point self-report (three questions) and the five BruxApp conditions (Table 1). Significant but weak correlations between possible AB were found only for the following BruxApp conditions: BA-relaxed (negative correlation), BA-bracing, BA-clenching and the total BA-AB score (positive correlations).

No correlations between AB and BA-teeth contact or BA-grinding could be observed.

### 3.4. Receiver Operating Characteristic (ROC) Curves

To examine the ability of the BruxApp application to assess subjects with and without possible AB (as defined by single-point self-response), ROC curves were calculated for each of the BruxApp conditions, as well as for the BA-AB variable (Table 2). The BA-relaxed condition was calculated separately, as its score had to be reversed.

The BruxApp conditions that had a significant ability to assess AB-positive and AB-negative subjects were BA-relaxed (AUC of 0.657, *p* < 0.01, 95% CI of 0.535–0.779), BA-bracing (AUC of 0.674, *p* < 0.01. 95% CI of 0.568–0.780), and BA-AB (AUC of 0.665, *p* < 0.01, 95% CI of 0.551–0.719).

The ability of BA-clenching to assess AB-positive and AB-negative subjects was low but approaching significance (AUC of 0.604, *p* = 0.08). The conditions of BA-teeth contact and BA-grinding had no predictive power whatsoever, with results no better than random guessing.

## 4. Discussion

An international expert panel suggested that both non-instrumental and instrumental approaches could be employed to assess bruxism. The clinical modes of assessment should be accurate (reliable and valid), applicable (feasible), affordable (cost-effective), and accessible (suitable for everyday clinical use) [1,22,25].

Non-instrumental approaches for bruxism assessment are mostly based on self-report (e.g., questionnaires, oral history, clinical examinations, and diaries), which are useful in gathering information on perceived bruxism activities but cannot quantify the duration and intensity of the muscle activity [1,16]. Additional information can be gathered by an evaluation of clinical data that are directly related to bruxism (i.e., signs), as well as others that are indirectly associated (i.e., symptoms and other potential consequences). It includes a clinical assessment of the temporomandibular joint (TMJ) and muscles status; an intraoral examination and inspection of the soft and hard tissues; dental-related signs of events potentially due to bruxism (e.g., signs in oral mucosal tissues, mechanical tooth wear, and broken restorations). The presence of such signs and symptoms may increase the level of bruxism assessment (from possible to probable bruxism), but is neither feasible nor affordable for wide population screenings [1].

Instrumental approaches, such as EMG recordings of sleep muscle activity, are considered as the gold standard for the assessment of sleep bruxism, but are difficult to apply during daytime to assess AB. Fujisawa et al. [31] showed that a small EMG (similar in size to a hearing aid) with voice recording was able to reveal specific EMG findings among subjects with self-reported daytime clenching compared to subjects without self-reported clenching. In a recent review by Yamaguchi et al. [32], the authors concluded that modern ultra-miniaturized wearable EMG devices enable precise performance equivalent to that of conventional stationary EMG devices. However, while reliable and valid, the use of EMG during daytime is not highly accessible and bears significant financial costs.

A recently presented EMA with the use of a designated smartphone application is a semi-instrumental approach for AB assessment, which enables subjects to recognize their habits and monitor changes over time [20,21,22,23].

To date, the use of single-point self-report is the most feasible and widely used approach to assess possible AB [1,29,33]. It has been widely used and quoted in numerous studies worldwide [24,25,26,27,28,29,34,35,36,37,38].

In the present study, the prevalence of AB among Israeli dental students, as defined by single-point self-report, was 34%, which is in full accordance with previous findings among Israeli adolescents (34.5%) [28]. Both studies used similar questions to define AB (two questions in the adolescent study and three questions in the present study). In a recent study carried out in Israel and Poland during the first stage of the COVID-19 pandemic, the prevalence of AB as assessed by similar single-point self-report questions, was around 30% (21% for males and 38% for females) for subjects 18–35 of age [39].

The non-instrumental approaches for assessing bruxism, such as single-point self-report, were sometimes presented as having poor concordance with instrumental approaches [30], partly due to a lack of consensus regarding the questions and the scoring system [23,29]. A study by Paesani et al. [14] found a strong positive correlation between self-reported and clinically based approaches to the assessment of awake clenching. The consistent findings regarding the prevalence of AB among young Israeli adults indicate that the use of single-point self-report to assess AB is reliable when the questions used and the scoring system are consistent.

The semi-instrumental EMA approach, via a smartphone app, is a relatively new strategy to evaluate AB, which is applicable, affordable, and accessible. Its validity and reliability still have to be determined. Manfredini et al. noted that even if close in time to the experience, the subjectivity of self-report may introduce some intra- and inter-individual reliability bias [21,40].

The present study population was a relatively homogenous group of young adults. The group was age- and education-wise similar to the populations studied by Bracci et al. [20] and Zani et al. [22] and showed resembling results. The mean frequencies of the conditions, as recorded by the EMA smartphone application, are resembling those reported by Zani et al. [22] during their first assessment and those reported by Bracci et al. [20] (Table 3). The most frequent behavior in the three studies was BA-relaxed (ca. 60–71%), followed by BA-teeth contact (ca. 14–20%) and BA-bracing (ca. 10–14%) [20,22]. Thus, teeth contact and jaw bracing/clenching are not unusual activities in otherwise healthy people, while grinding may be irrelevant to the healthy young adult population.

Bracci et al. also showed a low coefficient of intra-individual variability for the absence of AB over one week, namely, for the relaxed muscle condition [20]. Similarly, in the present study, the condition, which showed the lowest variability over the seven-day period of measurement, was the AB-relaxed condition.

The present results show that out of the five possible conditions reported by the application, only three (BA-relaxed, BA-bracing, and BA-clenching) correlated with the definition of AB as defined by single-point self-report. Moreover, only the BA-relaxed and BA-bracing conditions (as well as the combined variable BA-AB) presented an acceptable ability to assess AB-positive and AB-negative subjects. It is noteworthy that single-point self-report stands for possible AB and does not serve as a gold standard reference for AB. Nevertheless, the consistent findings among different Israeli populations grant the measure some validity.

The lack of correlation between self-reported AB and BA-teeth contact or BA-grinding is not surprising. Recent definition of bruxism specified the different muscle activities during wakefulness, which are associated with AB and removed grinding from the definition [1]. Teeth contact may be associated with regular oro-motor activities such as swallowing rather than with AB [15,27].

It is important to point out that besides recording real-time information about jaw position and/or muscular strain, being asked about a certain behavior in close contextual and temporal proximity to its occurrence increases self-awareness and may promote control and lead to cognitive change [22,40]. By doing so, a smartphone EMA application can serve as a biofeedback ecological momentary intervention (EMI) [41,42,43,44,45]. Zani et al. used the smartphone EMA assessment to monitor AB behaviors over time, through collecting data for two distinct periods with a one-month interval between them [22]. Their results showed an increase in the average frequency of relaxed jaw muscles reported in the study population in the second evaluation period. The authors suggested that the application may have a potential therapeutic use in myofascial pain patients with a self-reported history of AB [22].

The consensus obtained in 2018 [1] introduced the concept that muscle activity associated with AB (and/or SB) should not be considered as a disorder in ”otherwise healthy people,” but rather as a behavior that can be a risk factor for certain clinical conditions [1].

Little has been written on AB as a “normal behavior.” The assessment of such a normality is challenging, especially as the recommendation was to assess the activity in its continuum [1,22,40] rather than to determine clear cut-off points. EMA methods via smartphone applications have the potential to record muscle activity and to increase our knowledge on the epidemiological features of AB by studying the natural course and fluctuations of various AB behaviors [22].

The recent findings, which present recordings of AB behaviors via the use of a smartphone application [20,22], indicate that the frequency of the condition of relaxed muscles (BA-relaxed), among young adults, ranges from 60 to 70%. Additionally, this condition consistently shows the lowest CV rate over seven days of measurement. Therefore, this seems to be the most important condition in the assessment of AB. We suggest to re-consider the conditions of BA-teeth contact and BA-grinding while using methods such as EMA to assess AB. Minimizing the reported conditions to BA-relaxed versus a one or two other conditions describing muscular strain can make the application simpler. Future studies and discussions should be carried out to reach a consensus regarding the questions used to assess single-point self-report AB and the conditions recorded by EMA applications [46].

No study is without limitations. In the present study, AB was assessed by single-point self-report and was not objectively confirmed through an instrumental tool such as EMG. Therefore, the semi-instrumental EMA assessment tool via smartphone application was compared to possible AB rather than to probable (or even definite) AB. Comparing the application to probable AB, as defined by additional positive clinical inspection (or even to definite AB, as defined by daytime EMG), is an effort worthwhile trying. Additionally, the present report refers to a young, relatively homogeneous student population. Wider studies with more diverse populations are necessary to generalize the findings.

## 5. Conclusions

Definite assessment of AB is problematic due to the difficulty to use continuous EMG recordings during daytime activities. In the present study, two conditions (relaxed jaw muscles and muscle bracing), recorded by a designated EMA smartphone application, showed an acceptable ability to differentiate between subjects with and without possible AB, as defined by a single-point self-report. A combination of single-point self-report with a multi-point semi-instrumental EMA evaluation might be a proper approach to increase the level of AB assessment.

## Figures and Tables

**Figure 1 jcm-10-01699-f001:**
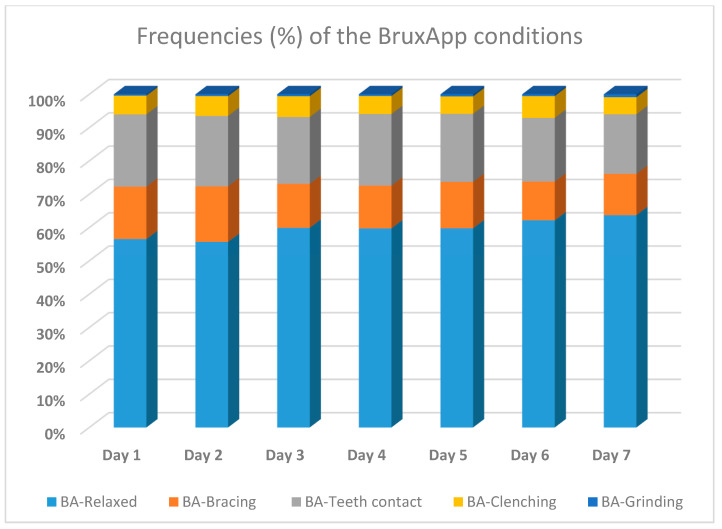
Frequencies of the BruxApp conditions per seven days of report.

**Table 1 jcm-10-01699-t001:** Correlations between awake bruxism (AB) as defined by single-point self-report and BruxApp conditions.

BruxApp	BA-Relaxed	BA-Bracing	BA-Teeth Contact	BA-Clenching	BA-Grinding	BA-AB
Pearson’s Correlation	−0.297	0.220	0.148	0.225	0.111	0.320
Sig. (2-tailed)	0.002	0.023	0.130	0.021	0.259	0.001
Number	106	106	106	106	106	106

**Table 2 jcm-10-01699-t002:** Summary of the ability of the BruxApp conditions to discriminate between patients with and without possible AB (ROC-AUC)).

Test Result Variable(s)	ROC-AUC	Std. Error	Asymptotic Sig. *	Asymptotic 95% Confidence Interval	
				Lower Bound	Upper Bound
BA-AB	0.665	0.058	0.006	0.551	0.779
BA-bracing	0.674	0.054	0.003	0.568	0.780
BA-teeth contact	0.556	0.059	0.349	0.439	0.672
BA-clenching	0.604	0.060	0.081	0.485	0.722
BA-grinding	0.505	0.060	0.936	0.387	0.623
BA-Relaxed **	0.657	0.062	0.008	0.535	0.779

* Significant results; ** the score of “BA-relaxed” was reversed.

**Table 3 jcm-10-01699-t003:** Frequencies of the ecological momentary assessment (EMA) conditions (%)—comparison among studies.

	Present Study	Zani et al. * [22]	Bracci et al. [20]
BA-relaxed	59.7	62	71
BA-clenching	5.7	3	3.7
BA-teeth contact	20.2	20	14
BA-grinding	0.6	1	0.1
BA-bracing	13.7	14	10 **

* During the first recording period; ** defined as bracing/jaw clenching.

## Data Availability

Additional data are available upon request from the first author (A.E.-P.).

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
