# Peer review of "Awake Bruxism—Single-Point Self-Report versus Ecological Momentary Assessment"

_jcm, 2021, doi:10.3390/jcm10081699_

Round 1

Reviewer 1 Report

The article evaluates Awake Bruxism by means of self-report and semi-instrumental approach – mobile application. The topic seems interesting. I have some recommendations regarding the content of article.

Introduction

Please add information on aetiology of awake bruxism.

Aim should be rewritten in a more concise form.

Materials and methods

Awake bruxism was not objectively confirmed – that should be emphasised in discussion or introduction.

Please describe the fundamentals of the application evaluation.

How was BA-Teeth contact evaluation performed? Who was using articulating paper?

Results

Try to present results as figures.

ROC curves could be omitted. Short description is adequate.

Discussion

Discuss other methods to determine awake bruxism.

Please add the limitations of the study.

Conclusions

Conclusions should answer the aim. Please thoroughly rewrite conclusions.

Author Response

BruxApp Paper (JCM)

Response to Reviewer 1:

Dear Reviewer

Thank you for taking the time to review the manuscript and for adding your valuable comments and recommendations.

We made all the corrections as proposed and we hope you will find that the manuscript has improved.

Introduction

Please add information on aetiology of awake bruxism. Response: Information about bruxism etiology was added to the introduction

Aim should be rewritten in a more concise form. Response: Aim was rewritten

Materials and methods

Awake bruxism was not objectively confirmed – that should be emphasized in discussion or introduction. Response: The fact that AB was not objectively confirmed and that single point self-report of AB enables only the confirmation of Possible AB was added to the Discussion and to the limitation sections.

Please describe the fundamentals of the application evaluation. Response: Basic description of the EMA fundamentals was added including reference to the extensive review by Shiffman et al (2008) 

How was BA-Teeth contact evaluation performed? Who was using articulating paper? Response: The articulating paper was used as a describing measure. The assumption was that participants (all dental students in the clinical phase of their studies) were well familiar with the articulating paper procedure and the explanation will enable them to better discriminate between Teeth contact and the condition Teeth clenching. Explanation was added to the text

Results

Try to present results as figures. Response: A figure of the “Frequencies of the BruxApp conditions” was added

ROC curves could be omitted. Short description is adequate. Response: ROC curves were omitted

Discussion

Discuss other methods to determine awake bruxism. Response: Methods to determine AB were described at the beginning of the Discussion

Please add the limitations of the study. Response: A paragraph of study limitations was added at the end of Discussion

Conclusions

nclusions should answer the aim. Please thoroughly rewrite conclusions. Response: Conclusions were rewritten

Reviewer 2 Report

Awake bruxism is quite vague to define especially compared to sleep bruxism. The manuscript is interesting to expand understanding of AB by utilizing EMA which is rather new assessment in this field.

  1. The basic idea in this study is to compare an EMA assessment and single point self-report for AB. Although EMA is one of the definite AB assessment tools, the accuracy is obviously lower than that of EMG. So, weak point of study design was not the highest accuracy procedure was used to evaluate the diagnostic tool of AB. This is the limitation of the study, so it should be described in discussion.
  2. Recording period in a day is not clear. Please describe when to start and stop such as from 10 am to 9 pm. 
  3. What was the reason to make 20 alert sounds for each recording day? Too many checking trials may cause inspire self-recognition for their AB restriction. Interval time between reminder alerts is not informed in the manuscript. Are they assigned in a random fashion?
  4. For statistical analysis, data were reported as mean values of the 7-day span per each condition. In table 1, BA-Relaxed increased as recording day progressed. Is there any statistical significance in BA-Relaxed between day1,2 and day7? If so, it is not proper to analyze each parameter through 7 days as a whole. In that case, a learned effect may affect the results as a confounding factor.
  5. In correlation analysis, the author described that significant correlation were found in a couple of pairs. However, statistics implied merely “not uncorrelated”. Looking at low correlation coefficients, I would like to conclude “there are weak correlations”.
  6.  In the result of ROC curve, single-point self-report was utilized as a gold standard. Please keep in mind that this is only a “probable” parameter. Be careful not to highlight too much on this point.
  7. Minor points

    P2L12

     Starting with Arabic numerals should be avoided.

    P12L38

     Bracci etal. → Bracci et al.

    P3L36

     106(63.2% female) → 106(67 females)

    P5 middle

     To examine the ability of the BruxApp application to discriminate between patients with and without AB

     → To examine the ability of the BruxApp application to discriminate between subjects with and without AB

    P7L19

    when the questions used and the and scoring system are consistent

     → when the questions used and the and scoring system are consistent

    Ref#15

    List all the authors instead of et al.

Author Response

BruxApp Paper (JCM)

Response to Reviewer 2:

 Dear Reviewer

Thank you for taking the time to review the manuscript and for adding your valuable comments and recommendations.

We made all the corrections as proposed and we hope you will find that the manuscript has improved.

  1. The basic idea in this study is to compare an EMA assessment and single point self-report for AB. Although EMA is one of the definite AB assessment tools, the accuracy is obviously lower than that of EMG. So, weak point of study design was not the highest accuracy procedure was used to evaluate the diagnostic tool of AB. This is the limitation of the study, so it should be described in discussion. Response: The issue was acknowledged in the manuscript, including a paragraph of the study limitations.

  1. Recording period in a day is not clear. Please describe when to start and stop such as from 10 am to 9 pm. Response: Information regarding recording period  was added.

  1.  What was the reason to make 20 alert sounds for each recording day? Too many checking trials may cause inspire self-recognition for their AB restriction. Interval time between reminder alerts is not informed in the manuscript. Are they assigned in a random fashion? Response: The software was initially programmed to send 20 alerts at random intervals.  Colonna et al (2020),  performed a study aimed to evaluate exactly that – namely, subjects’ compliance to the application. Their conclusion was that 12 (out of 20) responded alerts resulted the best results. The issue was clarified in the text.

  1. For statistical analysis, data were reported as mean values of the 7-day span per each condition. In table 1, BA-Relaxed increased as recording day progressed. Is there any statistical significance in BA-Relaxed between day1,2 and day7? If so, it is not proper to analyze each parameter through 7 days as a whole. In that case, a learned effect may affect the results as a confounding factor. Response: We omitted the report of  the 7 day mean values (Table 1) and reported in the text only  the range of  the mean frequencies of each of the BruxApp behaviors over the 7 days of report

  1. In correlation analysis, the author described that significant correlation were found in a couple of pairs. However, statistics implied merely “not uncorrelated”. Looking at low correlation coefficients, I would like to conclude “there are weak correlations”. Response: Correlations were defined as “Significant weak correlations”

  1.  In the result of ROC curve, single-point self-report was utilized as a gold standard. Please keep in mind that this is only a “probable” parameter. Be careful not to highlight too much on this point. Response: The point that single self report represents only Possible AB was addressed in the text

  1. Minor points. Response: All minor points were addressed

P2L12

 Starting with Arabic numerals should be avoided.- Corrected

P12L38

 Bracci etal. → Bracci et al.- Corrected

P3L36

 106(63.2% female) → 106(67 females)- Corrected

P5 middle

 To examine the ability of the BruxApp application to discriminate between patients with and without AB

 → To examine the ability of the BruxApp application to discriminate between subjects with and without AB - Corrected

P7L19

when the questions used and the and scoring system are consistent

 → when the questions used and the and scoring system are consistent - Corrected

Ref#15

List all the authors instead of et al. - Corrected

Round 2

Reviewer 1 Report

The manuscript was extensively corrected.

However, I do suggest adding description of EMA application: how does it work, what are the limitations, etc.

The cited reference is 13 years old and probably the application was up-dated. Moreover, the article in not in open access format, so the reader cannot easily understand the methodology.

Author Response

Thank you for your second review. The text was corrected according to your suggestion.

The cited reference is 13 years old and probably the application was up-dated. Moreover, the article in not in open access format, so the reader cannot easily understand the methodology. Response: A paragraph describing the principles of EMA , including cons and drawbacks , was added to the Introduction.

Reviewer 2 Report

Fig1 added in the revised version is good. On the other hand, Table 1 shows the same results. So it would be better to eliminate Table 1.

Results

P8 “Significant weak correlations” → “Significant but weak correlations”

Discussion

P8

“Instrumental approaches, such as EMG recordings of sleep muscle activity, are considered as the gold standard for the 25 assessment of sleep bruxism but are practically non-applicable during daytime to assess AB.” Although I agree a non instrumental approach is feasible, this opinion is not true. EMG recording studies targeting daytime have been performed successfully.  For example, you can find a report showing that a portable EMG device with voice recording proved to discriminate between functional and parafunctional EMGs during daytime. (J Oral Rehabil 2013, doi: 10.1111/joor.12087).

Author Response

Thank you for your second review. The text was corrected according to your suggestion.

Fig1 added in the revised version is good. On the other hand, Table 1 shows the same results. So it would be better to eliminate Table 1. Response: Table 1 was eliminated

Results

P8 “Significant weak correlations” → “Significant but weak correlations” Response: Corrected

Discussion

P8

“Instrumental approaches, such as EMG recordings of sleep muscle activity, are considered as the gold standard for the 25 assessment of sleep bruxism but are practically non-applicable during daytime to assess AB.” Although I agree a non instrumental approach is feasible, this opinion is not true. EMG recording studies targeting daytime have been performed successfully.  For example, you can find a report showing that a portable EMG device with voice recording proved to discriminate between functional and parafunctional EMGs during daytime. (J Oral Rehabil 2013, doi: 10.1111/joor.12087).  Response: A paragraph concerning the use of portable EMG during daytime was added to the Discussion